# Effects of Ramadan on clinical visits at a primary health center in a Rohingya refugee camp in Bangladesh

**Taizo Sakata**[1,2]☯*, **Odgerel Chimed-Ochir**[2]☯, **Md Omar Sharif Ibne Hasan**[3], **Mototaka Inaba**[1], **Yasuhiko Kamiya**[4], **Yui Yumiya**[2], **Tatsuhiko Kubo**[2]

1 Peace Winds Japan, Jinseki-Kogencho, Hiroshima, Japan, 2 Department of Public Health and Health Policy, Graduate School of Biomedical and Health Sciences, Hiroshima University, Hiroshima, Japan, 3 Dhaka Community Medical College & Hospital, Dhaka, Bangladesh, 4 School of Tropical Medicine and Global Health, Nagasaki University, Nagasaki, Japan

☯ These authors contributed equally to this work.
* taizo_sakata@peace-winds.org

## Abstract

In 2017, hundreds of thousands of Rohingya fled Myanmar to refugee camps in Cox's Bazar in southeastern Bangladesh. Peace Winds Japan and Dhaka Community Hospital Trust have jointly operated a primary health center (PHC) at Camp 14 in this region since 2018. Our objective was to determine the effect of Ramadan on clinical visits at this PHC during 2020. We conducted a retrospective analysis of patient data, including diagnoses and use of medications, by review of medical records from January to December 2020. We compared the percentage of visits for five health events (acute respiratory infection [ARI], non-communicable disease [NCD], gastritis, female related issues, and injury) for three consecutive 31-day periods: before Ramadan period (BR-period), Ramadan period, and after Ramadan period (AR-period). There were 11,286 PHC visits during 2020. The rate of new patient visits was significantly lower during Ramadan than during the BR-period (aOR: 0.447; 95%CI: 0.356, 0.561; P < 0.001). The three periods had no significant differences in mean patient age, patient sex, or surgical procedures. The most noteworthy differences were fewer visits for NCD during Ramadan than during the BR-period (OR: 0.461; 95%CI: 0.307, 0.693; P < 0.0004); more visits for gastritis during Ramadan than during the BR-period (OR: 1.246; 95%CI: 1.106, 1.404; P = 0.0019); and more visits for ARI during Ramadan than during the AR-period (aOR: 1.842; 95%CI: 1.567, 2.164, P < 0.0001). The overall decrease in PHC visits during Ramadan relative to the BR-period may be because these refugees devoted more time to prayer and associated religious practices during Ramadan, and therefore had less time available for visiting the PHC. The sharp decline in PHC visits for NCD and several other health events during Ramadan relative to the BR-period highlights the need to provide improved care for refugees during this period.

**Data availability statement:** The data used in this study are available from Figshare at the following link: https://figshare.com/articles/dataset/27100090. The dataset contains anonymized health information from the DCHT/PWJ Primary Health Center. All interested researchers can access the data in the same manner as the authors.

**Funding:** The authors received no specific funding for this work. The clinic operation was funded by JAPAN PLATFORM (grant number: NA). Website: https://www.japanplatform.org/index.html The funders had no role in study design, data collection and analysis, decision to publish, or preparation of the manuscript.

**Competing interests:** The authors have declared that no competing interests exist.

## Introduction

As of January 2024, approximately 975,350 Rohingya refugees (Forcibly Displaced Myanmar Nationals, FDMNs) were registered in Bangladesh as part of the joint registration exercise of Bangladesh and the United Nations High Commission for Refugees (UNHCR). The refugees now reside in 33 extremely congested camps that are formally designated by the Government of Bangladesh, which are located in the Ukhiya and Teknaf Upazilas of Cox's Bazar district and on the island of Bhasan Char [1]. According to the 2022 Joint Response Plan (JRP), a total population of 1.46 million individuals in this region (which includes refugees and individuals in the host communities) are in desperate need of assistance [2]. Peace Winds Japan and Dhaka Community Hospital Trust (DCHT) have jointly run a primary health center (PHC) in Camp 14 (population: 34,170) since January 2018, utilizing funds from the Japanese Official Development Assistance (ODA), the DCHT, and elsewhere.

This PHC is one of two PHCs in Camp 14 and is the only one equipped with Basic Emergency Obstetric and Neonatal Care (BEmONC) services that operates 24 hours a day. It currently provides basic primary care, such as first aid, minor surgery, immunizations, and maternal and child health services, treating an average of 160 patients per day, approximately 13% of whom are from the neighboring host communities. Since January 2018, this PHC has provided 24-hour emergency obstetric care, handling an average of 15 facility-based deliveries and approximately 300 antenatal care (ANC) visits per month. The PHC operates in collaboration with the Refugee Relief & Repatriation Commissioner (RRRC) and the Camp-in-Charge (CiC), and the Bangladesh health sector assures compliance with routine standards of data reporting. Prior to the onset of operations in February 2020, five other healthcare providers in Camp 14, including a few fixed health posts and occasional mobile clinics operated by other organizations, were only able to offer irregular services due to funding shortages and insufficient financial support. These deficiencies likely decreased attendance at these clinics.

The Rohingya people are an ethnic group from Myanmar. Myanmar is a majority Buddhist state, but the Rohingya people are primarily Muslim, although a small number are Hindu. Millions of adult Muslims worldwide, including the majority of Rohingya, observe fasting during the holy month of Ramadan each year. This religious observance requires observers to refrain from eating food, drinking water or other beverages, and smoking from dawn until sunset for 29–30 days [3]. There are several groups who are not required to fast, such as pregnant or nursing women if there is reason to fear harm to the mother or child, prepubescent children, older people, those with mental illness, the sick, and those with chronic illness if there is a serious medical concern [4]. However, many exempt Muslims who have chronic medical conditions may still choose to fast [5]. In general, daytime activity levels during Ramadan tend to be lower, and observers devote more time to prayer and associated washing practices.

Although many studies have shown that social determinants, such as spirituality, religion, and personal beliefs, are associated with health-related behaviors and

treatments [6,7], no publications in professional journals have yet described the impact of Ramadan on medical care in the Rohingya refugee camps. Our objective was to determine the impact of the Ramadan month on clinical visits at our PHC in a Rohingya refugee camp.

## Materials and methods

### Ethics statement

The study primarily involved a retrospective analysis of data, which included data from participants under the age of 18. Given the retrospective nature of this study, retrospectively obtaining written informed consent from all guardians was not feasible. However, stringent measures were taken to ensure all data were anonymized and handled with the utmost confidentiality to prevent the identification of individual participant. The study was conducted in accordance with the Declaration of Helsinki, and the protocol, including the method of consent, was reviewed and approved by the Ethical Committee for Epidemiology of Hiroshima University (protocol code E-2745, dated 2022/1/22). Additional information regarding the ethical, cultural, and scientific considerations specific to inclusivity in global research is in the S1 Checklist.

### Study design

This study was a retrospective analysis of data collected in our PHC from January 2020 to December 2020.

### Study site

The refugees were located at different camps in Cox's Bazar district in southeastern Bangladesh. This area includes the world's largest and most crowded refugee camps, and there were 926,561 Rohingya refugees as of 31 March 2022 [8]. Our PHC provides services to individuals in Hakimpara (Camp 14) and to those in five host communities in the Whykong and Palongkhali Unions, which are located in the Ukhiya and Teknaf Upazilas of Cox's Bazar district. The total population (41,322) consists of 34,100 refugees in Camp 14 and 7,222 individuals in host communities (Table 1) [9,10]. The host community accounted for 17.4% of the total direct beneficiaries, and approximately 13% of patients who visited our PHC.

### Data collection

To assist with clinical monitoring and reporting to administrative authorities, a database was introduced at the PHC in March 2019. The data collected through this system were primarily intended for monitoring clinic activities and reporting to stakeholders, rather than for research or publication. This database uses a data collecting and recording system referred to as the Japanese Surveillance in Post Extreme Emergencies and Disasters (J-SPEED). J-SPEED was based on the Surveillance in Post Extreme Emergencies and Disasters (SPEED) in the Philippines, and was proposed by the Japanese

**Table 1. Age and sex of the refugee community in Camp 14 and in host communities.**

| Target Population | Refugees in Camp 14 | | | | | | Host Communities in Whykong (Lambaghona, Horikhola) and Palongkhali Union (Hakimpara, Jamtoli, Telkhola) | | | | | |
|---|---|---|---|---|---|---|---|---|---|---|---|---|
| | Male | | Female | | Total | | Male | | Female | | Total | |
| Age, years | # | % | # | % | # | % | # | % | # | % | # | % |
| 0-4 | 2692 | 7.9 | 2674 | 7.8 | 5366 | 15.7 | 679 | 9.4 | 672 | 9.3 | 1351 | 18.7 |
| 5-11 | 3920 | 11.5 | 3676 | 10.8 | 7596 | 22.3 | 816 | 11.3 | 773 | 10.7 | 1589 | 22.0 |
| 12-17 | 2654 | 7.8 | 2351 | 6.9 | 5005 | 14.7 | 498 | 6.9 | 491 | 6.8 | 989 | 13.7 |
| 18-59 | 6591 | 19.3 | 8054 | 23.6 | 14645 | 42.9 | 1560 | 21.6 | 1466 | 20.3 | 3026 | 41.9 |
| >60 | 827 | 2.4 | 661 | 1.9 | 1488 | 4.4 | 10 | 1.8 | 137 | 1.9 | 267 | 3.7 |
| Total | 16684 | 48.9 | 17416 | 51.1 | 34100 | 100 | 3683 | 51.0 | 3539 | 49.0 | 7222 | 100.0 |

Joint Committee for Disaster Medical Recording after the Great East Japan Earthquake in 2011. This tool emphasizes the use of standardized and simple reporting for Emergency Medical Teams (EMTs) during disasters, and addresses the necessity for prompt data collection and standardized reporting so that public emergency services can be provided in a swift and coordinated manner [11]. The J-SPEED form is a one-page document with 40–50 items in categories such as demographics, health events, and treatments. Physicians mark relevant items on the form, and data managers or nurses then enter these data into an Excel spreadsheet. The tool generates daily summary reports and provides raw data as comma-separated values (CSV) for subsequent detailed analysis.

The data collection utilized a customized form that was developed by referencing different disease categories in the J-SPEED form. This form was further modified based on input from local physicians, so that it better characterized conditions related to peri-natal care and the most common illnesses in the study area. A specially trained paramedic reviewed medical records and entered patient information, diagnoses, medications, procedures, and outcomes into the Excel spreadsheet. The data were collected daily and summarized in various reports: "Early Warning, Alert and Response System (EWARS) Reports" (weekly), "Who does What, Where, and When (4W) Reports" to the Health Sector (monthly), "Sexual and Reproductive Health (SRH) Reports" to the SRH sector (monthly), and "District Health Information System 2 (DHIS-2) Reports" to the Bangladesh Government (monthly). All Excel data were stored with password protection, and all personal information was analyzed in a confidential manner to prevent individual identification.

## Data analysis

Data were analyzed from January 2020 to December 2020, because data collected during this period were complete and comprehensive. A patient could have multiple diseases (health events) during PHC visits, so all health events were recorded. We selected three periods (31 days each): (*i*) the "before Ramadan period" (BR-period) was from March 23 to April 22, 2020; (*ii*) the "Ramadan period" was from April 23 to May 23, 2020, with exclusion of the Eid-ul-Fitr (Breaking of the Fast) holiday from May 24–26; and (*iii*) the "after Ramadan period" (AR-period) was from May 27 to June 26, 2020. We selected a 31-day window for three main reasons. First, a previous study indicated that a one-month period is suitable for examining Ramadan-related effects [12]. Second, using a comparable observation window before and after the intervention (equal in length to the intervention period itself) promotes symmetry in measurement, thereby simplifying the attribution and interpretation of observed changes [13]. Finally, shorter windows would reduce the total number of data points in the pre- and post-Ramadan segments, limiting statistical power, whereas very long windows risk diluting the specific Ramadan effect by introducing excessive baseline variability, including seasonal effects [13–15].

Patient characteristics, specific health events based on diagnosis, procedures (especially surgical procedures), and referral status as outcome were examined. Then, data from the Ramadan period were compared to BR-period and AR-period, because previous research identified differences in emergency department (ED) visits during these periods [12]. Data from the host community were excluded, because the host community includes individuals in the Buddhist Chakma tribe. Children younger than 15-years-old were also excluded because Ramadan is only obligatory for those who have reached puberty.

A total of 60 diseases and symptoms were registered as diagnoses. When data were initially gathered for monitoring purposes, disease names were not well organized and some entries were merely symptoms (e.g., weakness, headache, etc.) rather than specific diseases. Thus, patients with cold symptoms or with upper or lower respiratory tract infections were combined in the category of acute respiratory infection (ARI); patients with hypertension, chronic obstructive pulmonary disease, asthma, diabetes, and chronic bronchitis were combined in the category of non-communicable disease (NCD). Among all health events registered at PHC, ARI, NCD, digestive problem (gastritis or diarrhea), female-related issues (ante-natal care [ANC] and post-natal care [PNC]), and injury were selected for the analysis. ARI and NCD were chosen to represent common acute and chronic conditions, while injuries reflect urgent medical needs within the camp.

Gastritis was included due to evidence in the literature suggesting a link between Ramadan fasting and gastrointestinal symptoms [16]. ANC and PNC were selected due to ongoing concerns that traditional Rohingya cultural norms may limit women's access to maternal care, despite recommendations from WHO and the health cluster to improve maternal and newborn outcomes. Other conditions such as family planning and oral health were not prioritized, as they do not pose the same level of immediate clinical urgency as ARI, NCDs, and injuries. Other conditions, including general weakness, headache, and musculoskeletal problems, were often nonspecific complaints without clear diagnoses, and thus were excluded from the analysis.

In terms of statistical analysis, we first calculated a 7-day moving average of all PHC visits and health events during 2020, with a focus on the BR-period, Ramadan period, and AR-period. A descriptive analysis was also performed to examine the relationships of the reason for a health events with the type of visit (new or follow-up), demographic variables, and procedure (surgery) and outcome (referral) during each period.

In our analysis, we conducted separate multivariable logistic regression models, each using a binary independent variable representing the period of PHC visits (BR period vs. Ramadan period, and Ramadan period vs. AR period). The dependent variables analyzed included the type of visit (new or follow-up), procedures performed, patient outcomes, and specific health events such as ARI, NCDs, diarrhea, injuries, antenatal care (ANC), and postnatal care (PNC).

All regression models were adjusted for age and sex to account for potential confounding. Additionally, we tested for interaction effects between age and the Ramadan period as well as between sex and the Ramadan period, but found no significant interactions. As a result, we proceeded with models adjusted solely for age and sex, without including interaction terms.

Because there were no referrals during Ramadan, standard binary logistic regression was inappropriate for assessment of visit outcome. Therefore, Firth logistic regression was used to identify the association between Ramadan and patient referral. This technique considers the presence of binary independent variables in the dataset, because complete separation can result in estimates of infinite odds ratios. Firth logistic regression is a modified version of standard logistic regression that utilizes a penalized likelihood approach to eliminate bias from complete separation by adding a small constant to the diagonal of the observed information matrix. This approach helps to stabilize parameter estimates and provides more accurate results [17,18]. *STATA version 17.0 (STATA Corp; College Station, Texas USA) was used for data analysis.*

## Results

Fig 1 shows the 7-day moving average number of PHC visits and health events during 2020. During the BR-period, there was an average of 125 health events (111 PHC visits) per day. During the Ramadan period, there was an average of 106 health events (93 PHC visits) per day. During the AR-period, there was an average of 130 health events (116 PHC visits) per day.

Table 2 summarizes health events recorded at PHC according to sex, mean age, and age group of patients during four different time periods: all of 2020, the BR-period, the Ramadan period, and the AR-period. There were 11,116 health events recorded during the entire year, and fewer health events during the Ramadan period (3,391) than during the BR-period (4,019) and the AR-period (3,706). During each period, 72–74% of health events were by females and the mean age ranged from 30.5 to 33.1 years. Overall, 30–36% of visits were from the youngest group (15–24 years), 48–54% were from the second-youngest group (25–44 years), 10–18% were from the second-oldest group (45–64 years), and 2–3% were from the oldest group (>65 years).

Relative to the BR-period and AR-period, the Ramadan period had a greater percentage of patients in the youngest group (36%) and a smaller percentage of patients in the second-youngest group (48%). There was also an increasing percentage of patients in the second-oldest group during the progression from the BR-period (10%) to the Ramadan period (13%) and to the AR-period (18%). Pair-wise comparisons (Ramadan *vs.* BR-period and Ramadan *vs.* AR-period) showed

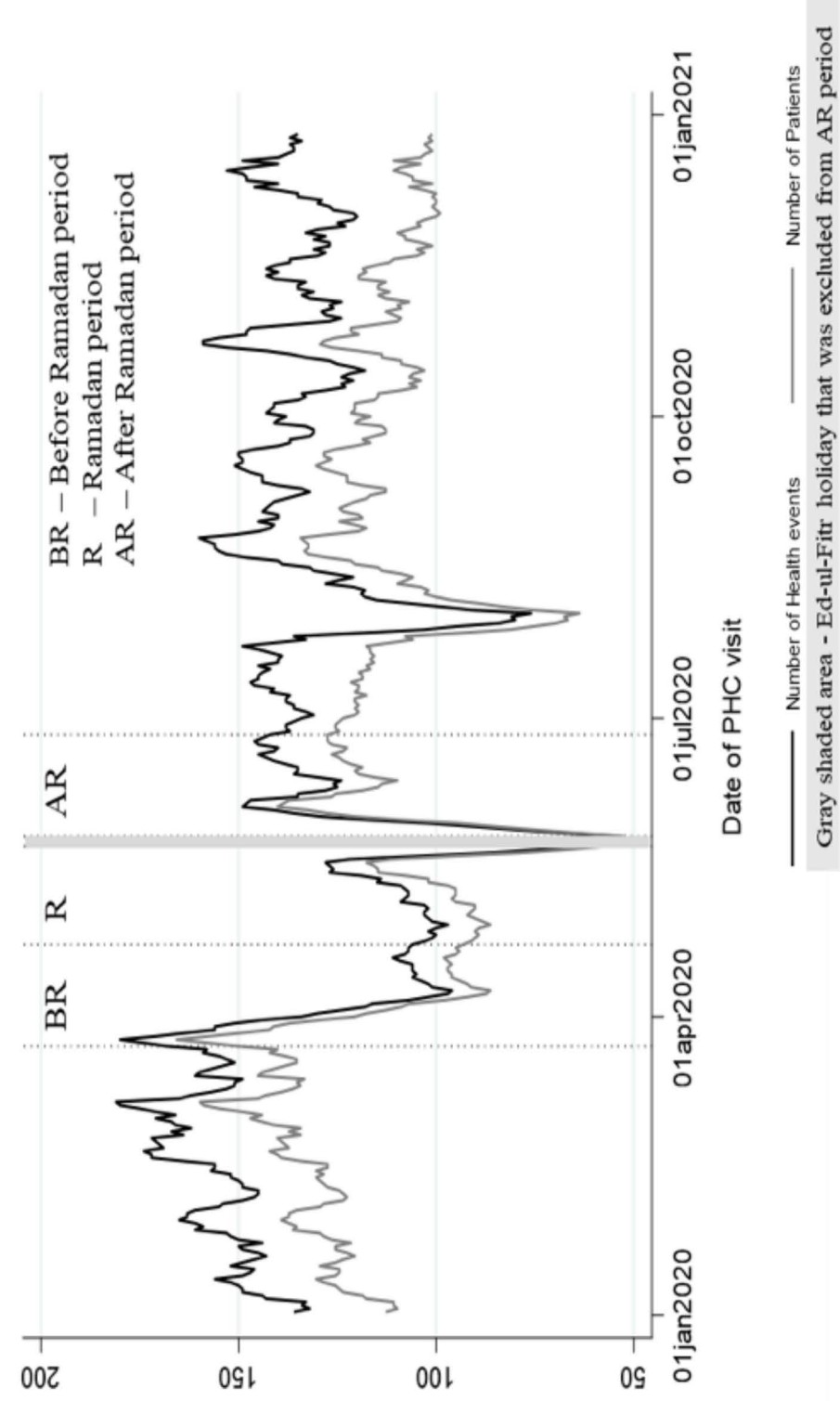

**Fig 1. Seven-day moving average number of PHC visits and health events during 2020.**

**Table 2. Age and sex of patients who visited the PHC during all of 2020, and during the BR-period, Ramadan period, and AR-period.**

| | Entire study period | | | Before Ramadan (BR) | | | Ramadan[1] (R) | | | After Ramadan (AR) | | | p value[2] | |
| | | | | 2020/03/23–2020/04/22 | | | 2020/04/23–2020/05/23 | | | 2020/05/27–2020/06/26 | | | R vs BR period | R vs AR period |
| | Total/Average number of health events (HEs) | | % of total HEs | Total/Average number of HEs | | % of total HEs | Total/Average number of HEs | | % of total HEs | Total/Average number of HEs | | % of total HEs | | |
|---|---|---|---|---|---|---|---|---|---|---|---|---|---|---|
| **Sex** | | | | | | | | | | | | | | |
| Male | 2,993 | 32 | 27% | 1,063 | 34 | 26% | 955 | 31 | 28% | 975 | 31 | 26% | 0.099 | 0.0826 |
| Female | 8,123 | 87 | 73% | 2,956 | 95 | 74% | 2,436 | 79 | 72% | 2731 | 88 | 74% | | |
| **Age, years** | | | | | | | | | | | | | | |
| Mean (SD) | 31.6 (12.9) | | | 30.5 (12.1) | | | 31.2 (13.1) | | | 33.1 (13.6) | | | | |
| Range | | | | | | | | | | | | | | |
| 15-24 | 3,657 | 39 | 33% | 1345 | 43 | 33% | 1215 | 39 | 36% | 1097 | 35 | 30% | <.0001 | <.0001 |
| 25-44 | 5,666 | 61 | 51% | 2180 | 70 | 54% | 1643 | 53 | 48% | 1843 | 59 | 50% | | |
| 45-64 | 1,504 | 16 | 14% | 405 | 13 | 10% | 437 | 14 | 13% | 662 | 21 | 18% | | |
| >65 | 289 | 3 | 3% | 89 | 3 | 2% | 96 | 3 | 3% | 104 | 3 | 3% | | |
| Total | **11,116** | | 100.0% | **4,019** | 130 | 100.0% | **3,391** | 109 | 100.0% | **3,706** | 120 | 100.0% | 0.0343 | 0.0705 |

SD: Standard Deviation

[1]The Ed-ul-Fitr holiday (May 24–26) was excluded from this analysis.

[2]P-values were from a chi-square test that compared health events for the different sexes and age groups, and from a one-sample independent *t*-test that compared the average events in the clinic during Ramadan with the BR-period and with the AR-period.

there were significant differences in the number of health events among age groups in each comparison (both P < 0.0001) and a significant difference in total health events between the BR-period and Ramadan period (p = 0.0343).

Table 3 shows the number and types of health events, procedure, and outcome of visits for all of 2020, and for the BR-period, Ramadan period, and AR-period. Relative to the BR-period, the Ramadan period had a smaller percentage of new visits (3.3% *vs.* 6.8%, P < 0.001), a smaller percentage of NCD (1.2% *vs.* 2.1%, P = 0.005), a larger percentage of gastritis (20.0% *vs.* 16.5%, P < 0.001), a smaller percentage of diarrhea (0.7% *vs.* 1.4%, P = 0.004), a smaller number of ante-natal care (8.9% *vs.* 11.5%, P = 0.0002), a larger percentage of visits for post-natal care (0.5% *vs.* 0.1%, P = 0.002), and a smaller percentage of injury (0.8% *vs.* 1.7%, P < 0.0002). Relative to the AR-period, the Ramadan period had a smaller percentage of new visits (3.3% *vs.* 5.2%, P < 0.0001) and a larger percentage of ARI (12.3% *vs.* 7.0%, P < 0.0001).

We then extended this analysis by performing multivariable logistic regression analysis, with adjustment for age and sex, to calculate adjusted odds ratios (aORs) (Table 4). Relative to the BR-period, the Ramadan period had a lower odds for new visits (aOR: 0.447; 95%CI: 0.356, 0.561), a lower odds for NCD (aOR: 0.461; 95%CI: 0.307, 0.693), a greater odds for gastritis (aOR: 1.246; 95%CI:1.106, 1.404), a lower odds for diarrhea (aOR: 0.499; 95%CI: 0.306, 0.814), a lower odds for ante-natal care (aOR: 0.777; 95%CI: 0.662, 0.911), a greater odds for post-natal care (aOR: 4.453; 95%CI: 1.651, 12.105), and a lower odds for injury (aOR: 0.434; 95%CI: 0.276, 0.683). Relative to the AR-period, the Ramadan period had a lower odds for new visits (aOR: 0.610; 95%CI: 0.479, 0.775) and a greater odds for ARI visits (aOR: 1.818; 95%CI: 1.544, 2.140).

## Discussion

This study aimed to determine the effect of Ramadan on clinical visits and specific health events at PHC in a Rohingya refugee camp. We identified that relative to the BR-period, the Ramadan period had fewer new visits, NCD, diarrhea,

PLOS Global Public Health

**Table 3. New visits, types of health events, procedure, and outcome of health events during all of 2020, and during the BR-period, Ramadan period, and AR-period.**

| | Entire study period | | | Before Ramadan (BR) 2020/03/23–2020/04/22 | | | Ramadan (R)[1] 2020/04/23–2020/05/23 | | | After Ramadan (AR) 2020/05/27–2020/06/26 | | | p value[2] | |
|---|---|---|---|---|---|---|---|---|---|---|---|---|---|---|
| | Total N of health events (HEs) | | % of total HEs | Total N of HEs | | % of HEs attended at clinic | Total N of HEs | | % of HEs attended at clinic | Total N of HEs | | % of HEs attended at clinic | R vs BR period | R vs AR period |
| | Yes | No | | Yes | No | | Yes | No | | Yes | No | | | |
| **Characteristics of PHC visit** | | | | | | | | | | | | | | |
| New visit (Ref: follow-up) | 576 | 10,702 | 5.1% | 272 | 3,747 | 6.8% | 112 | 3,279 | 3.3% | 192 | 3,514 | 5.2% | <.0001 | 0.0001 |
| **Type of health event** | | | | | | | | | | | | | | |
| Acute respiratory infection | 1,168 | 9,948 | 10.3% | 489 | 3,530 | 12.2% | 418 | 2,973 | 12.3% | 261 | 3,445 | 7.0% | 0.8588 | <.0001 |
| Non-communicable diseases | 163 | 10,953 | 1.4% | 83 | 3,936 | 2.1% | 41 | 3,350 | 1.2% | 39 | 3,667 | 1.1% | 0.005 | 0.574 |
| Digestive problem | | | | | | | | | | | | | | |
| Gastritis | 2,077 | 9,039 | 18.4% | 663 | 3,356 | 16.5% | 678 | 2,713 | 20.0% | 736 | 2,970 | 19.9% | 0.0001 | 0.905 |
| Diarrhea | 102 | 11,014 | 0.9% | 55 | 3,964 | 1.4% | 23 | 3,368 | 0.7% | 24 | 3,682 | 0.6% | 0.004 | 0.771 |
| Female-related issue | | | | | | | | | | | | | | |
| Ante-natal care | 1,077 | 10,039 | 9.5% | 462 | 3,557 | 11.5% | 301 | 3,090 | 8.9% | 314 | 3,392 | 8.5% | 0.0002 | 0.555 |
| Post-natal care | 32 | 11,084 | 0.3% | 5 | 4,014 | 0.1% | 18 | 3,373 | 0.5% | 9 | 3,697 | 0.2% | 0.0025 | 0.055 |
| Injury | 139 | 10,977 | 1.2% | 70 | 3,949 | 1.7% | 26 | 3,365 | 0.8% | 43 | 3,663 | 1.2% | 0.0002 | 0.115 |
| **Procedure** | | | | | | | | | | | | | | |
| Surgical procedure | 273 | 10,851 | 2.4% | 95 | 3,924 | 2.4% | 76 | 3,315 | 2.2% | 94 | 3,612 | 2.5% | 0.756 | 0.438 |
| | | – | | | | 0.0% | | | 0.0% | | | 0.0% | | |
| **Outcome of visit** | | | | | | | | | | | | | | |
| Referral[3] | – | 11,065 | 0.0% | – | 4,019 | 0.0% | – | 3,391 | 0.0% | – | 3,655 | 0.0% | NA | NA |

[1]The Ed-ul-Fitr holiday (May 24–26) was excluded from this analysis.

[2]P values are from the Chi square test or Fisher's test.

[3]No referral data were available for 51 cases.

ANC, and injury, but more ARI, gastritis, and PNC. Relative to the AR-period, the Ramadan period had significantly fewer new visits and significantly more ARI.

Finding of the decrease of new visits during Ramadan aligns with the results of Khojah et al., who observed a decrease in emergency department visits during Ramadan in a Muslim-majority country [19]. Studies from other Muslim-majority countries have predominantly reported neutral or decreased emergency room utilization during Ramadan, highlighting variability in healthcare-seeking patterns across different contexts [20]. Şimşek et al. suggested that discrepancies in findings across studies may be attributed to differences in fasting rates or regional variations in cultural practices [12]. The observed increase in visits during the BR period may reflect preparatory behavior among patients. These behaviors could include confirming health status, completing necessary treatments, and optimizing medications to mitigate potential health risks during fasting. However, it is important to note that there are no universally

**Table 4. Multivariable logistic regression analysis of the relationship of the date of a PHC visit with new visits, types of health events, procedure, and outcome of visits.**

| Variables | Ramadan vs Before Ramadan period | | | Ramadan vs After Ramadan period | | |
|---|---|---|---|---|---|---|
| | AOR | 95% CI | P-value | AOR | 95% CI | P-value |
| **Demographic characteristics** | | | | | | |
| Sex (Ref: Male) | 0.944 | 0.850-1.049 | 0.285 | 0.843 | 0.757-0.939 | 0.0019 |
| Age[1] | 0.994 | 0.910-1.085 | 0.886 | 1.368 | 1.252-1.494 | <0.0001 |
| **Character of health event** | | | | | | |
| New visit (Ref: Follow-up) | 0.447 | 0.356-0.561 | <0.0001 | 0.610 | 0.479-0.775 | <0.0001 |
| **Type of health events** (Ref: Absence of disease) | | | | | | |
| Acute respiratory infection | 1.017 | 0.884-1.169 | 0.818 | 1.818 | 1.544-2.140 | <0.0001 |
| Non-communicable diseases | 0.461 | 0.307-0.693 | 0.0002 | 1.289 | 0.819-2.027 | 0.2728 |
| Digestive problem | | | | | | |
| Gastritis | 1.246 | 1.106-1.404 | 0.0003 | 1.042 | 0.927-1.172 | 0.4883 |
| Diarrrhea | 0.499 | 0.306-0.814 | 0.0053 | 1.085 | 0.610-1.930 | 0.7808 |
| Female-related issue | | | | | | |
| Ante-natal care | 0.777 | 0.662-0.911 | 0.0019 | 0.975 | 0.820-1.160 | 0.7758 |
| Post natal care | 4.453 | 1.651-12.015 | 0.0032 | 2.126 | 0.952-4.748 | 0.0658 |
| Injury | 0.434 | 0.276-0.683 | 0.0003 | 0.654 | 0.400-1.068 | 0.0897 |
| **Outcome of the visits** | | | | | | |
| Surgical procedure (Ref: No surgical procedure) | 0.941 | 0.694-1.278 | 0.690 | 0.851 | 0.626-1.157 | 0.304 |
| Referral[2] (Ref: No referral) | 0.353 | 0.014-8.678 | 0.524 | 0.302 | 0.012-7.43 | 0.464 |

Abbreviations: N, number; CI, confidence interval; aOR, adjusted odds ratio.

[1]Ordinal regression.

[2]Firth logistic regression.

accepted cut-off points for defining the BR and AR periods in such studies, which introduces a degree of subjectivity in determining these timeframes. We believe that the most suitable analytical approach to compare the three periods without relying on specific cut-off points would be interrupted time series analysis. However, our dataset contains too few event-specific data points to apply this method effectively. Further research, such as patient interviews or qualitative studies, would be required to explore this phenomenon and confirm its relevance to visit patterns. In addition, lower activity levels during Ramadan, with more time spent in prayer and associated washing practices, may be responsible for fewer PHC visits during Ramadan, because it takes about 40 min to walk from the furthest block in Camp 14 to our PHC. The Joint Multi-Sector Needs Assessment (J-MSNA) of 2019 also examined this issue [21]. This previous study showed that for the 3% of individuals who reported having an illness serious enough to require medical treatment but who did not seek treatment, the most-frequently reported reasons for not seeking treatment were: (*i*) overcrowded health services (long wait times); (*ii*) no availability of treatment; (*iii*) poor behavior of health service staff; and (*iv*) need for long-distance travel or lack of transportation.

The fluctuations in the number of PHC visits that we observed throughout the year reflect a complex interplay of various factors. In 2020, the onset of the COVID-19 pandemic led to significant societal changes and a decrease in outpatient visits. Internal reports indicated that events such as the installation of barbed wire around the refugee camp, heightened security concerns due to fires, and certain other incidents also contributed to these fluctuations. Additionally, changes in the availability of local healthcare services may have impacted visits at our PHC. These many factors explain the observed variability in the number of PHC visits throughout the year, and make it challenging to identify specific reasons for each peak and trough.

Despite the overall decrease in PHC visits during the Ramadan period, the proportion of visits requiring surgical procedures remained relatively unchanged, indicating that Ramadan had no effect on urgent medical needs, such as injuries and other conditions requiring immediate attention. However, there were fewer injuries during Ramadan compared to the BR-period, possibly because there were decreased physical activities and outdoor work during fasting hours.

The probability of ARI was greater during Ramadan compared to the AR-period, but there was no significant change from the BR-period to the Ramadan period. This could be due to seasonal and environmental factors that may have influenced the fluctuation of ARI cases post-Ramadan. Additionally, while fasting during Ramadan could potentially alter immune function or activity levels, leading to higher susceptibility to infections, further studies are needed to determine the extent of these effects in the refugee population. On the contrary, visits for NCD are generally not urgent, and there were fewer NCD visits during Ramadan. In some cases, religious beliefs may discourage patients from undergoing blood sampling during fasting hours, which could in fluence their willingness to seek care, particularly for conditions such as diabetes that typically requires blood tests. However, as our PHC offers only minimal laboratory services and most diagnoses are made clinically, the actual impact of this factor in our setting is likely limited. As in other countries, the prevention and control of NCDs are addressed by UN agencies and humanitarian organizations in the Rohingya refugee camps. The BRAC international development organization conducted a needs assessment by analysis of 400 Rohingya households in 8 camps during March 2018, and reported that 37.3% of these individuals had hypertension [22]. A recent systematic review and meta-analysis showed that fasting during Ramadan was not associated with acute cardiovascular events [23]. However, noncompliant hypertensive patients and individuals with unstable angina, decompensated heart failure, recent cardiac intervention or surgery, or recent myocardial infarction should follow medical advice and refrain from fasting during Ramadan [24]. The International Diabetes Federation (IDF) and the Diabetes and Ramadan (DAR) International Alliance created comprehensive guidelines for management of fasting patients with type 1 and type 2 diabetes [25]. They considered religious and medical issues, and recommended when a diabetic patient should or should not fast according to different risk categories. For these reasons, the decrease in PHC visits during Ramadan could have a negative impact on the health of the Rohingya population, and medical practitioners should provide appropriate medical care and advice before, during, and after Ramadan.

Our study found significantly more gastritis during the Ramadan period (20%) than during the BR-period (16.5%). In contrast, Keshteli et al. found no relationship between the frequency of gastrointestinal symptoms and Ramadan fasting, except that constipation increased significantly after Ramadan fasting (OR: 1.99; 95%CI: 1.05, 3.80; P<0.05) [26]. Another study reported that symptoms of dyspepsia, including bloating, indigestion, and heartburn, were common in fasting patients, especially those who had unhealthy eating habits, such as excessive eating during the iftar or suhoor meals [19]. This suggests that patients should be advised to avoid excessive eating and consumption of foods that can trigger these symptoms. Our data do not provide details regarding the types of gastrointestinal problems, so further research is needed to determine the causes of the increased gastritis during Ramadan.

The Rohingya society is patriarchal and embraces conservative norms that limit the access of women and girls to careers, education, medical information, and public services. Because decision-making is mostly by men, over 60% of married women in the refugee and host communities require permission to access health services [27]. In this study, we observed that the percentage of PHC visits by females remained relatively steady, with 74% during the BR-period, 72% during the Ramadan period, and 74% during the AR-period. However, there were significant changes in the rates of visits for ANC and PNC. ANC visits decreased significantly from 11.5% during the BR-period to 8.9% during the Ramadan period. This decrease may reflect the challenges and restrictions faced by pregnant women in accessing healthcare during Ramadan. Conversely, PNC visits increased from 0.1% during the BR-period to 0.5% during the Ramadan period. There may be multiple reasons for this increase. One plausible explanation is the heightened medical needs of mothers for their newborns immediately postpartum, which may drive them to seek medical attention despite the influence of these cultural and religious practices. In our PHC, two local Rohingya traditional birth attendants were employed and

actively promoted the importance of PNC, particularly to women who delivered at our facility. Another factor could be the distribution of non-food items that were provided as incentives for these women to visit the PHC, which may increase the frequency of PNC visits. As PNC services were primarily offered to women who gave birth at our PHC, the observed increase may reflect the effectiveness of these targeted initiatives rather than a broader shift in health-seeking behavior during Ramadan. Additionally, the variability in the number of births from month to month can introduce bias, and the changes in the percentages of visits for ANC and PNC may not be directly related to Ramadan. More efforts are required to address the SRH needs of women and girls in the Rohingya refugee camps, because lack of access to SRH services is often unreported. A significant challenge is that many pregnant women in these camps do not receive antenatal and post-natal check-ups at the frequency recommended by the WHO. Moreover, there is a critical shortage of 24/7 health centers that provide emergency obstetric and newborn care, and this can lead to delays in the management of complications [28]. Therefore, promoting deliveries in PHCs and encouraging women to seek antenatal and postnatal services — even during Ramadan — is crucial to improving the outcomes of mothers and their children.

The observed shifts in health-seeking behavior and disease patterns, shaped by cultural and religious practices, highlight the need for targeted interventions in the Rohingya refugee camps. Tailoring healthcare strategies to these patterns, such as optimizing staff allocation and adjusting medical resources, can improve healthcare delivery, even with limited resources. Future research should focus on understanding and addressing these cultural factors across the entire camp to better inform sector-wide health strategies.

## Limitation

This study has several limitations. First, the data were collected primarily for clinical monitoring and administrative reporting purposes, rather than for research, which may limit the depth of analysis. As a result, the study design and data collection methods were not optimized for research purposes, which may limit the depth of analysis. Secondly, many diagnoses were based on interviews and physical examinations without the use of diagnostic tests that are commonly available in urban healthcare settings, and this may limit the accuracy of the diagnoses. Thirdly, the study period coincided with the global COVID-19 pandemic, and this may have influenced healthcare-seeking behaviors and access to health services in ways not fully accounted for by our analysis. Although this study provides valuable quantitative insights, it lacks qualitative data that could enhance the interpretation and meaning of the results. Lastly, it should be noted that this study should not be considered representative of all PHCs in the Rohingya refugee camps.

## Conclusion

In conclusion, the number of visits at our PHC in a Rohingya refugee camp decreased significantly from the BR-period to the Ramadan period, particularly visits by new patients, and by those needing care for NCD, diarrhea, ANC and injury. Despite this overall decline in PHC visits, there were only minor changes in the rate of acute health events, sex distribution, age, and surgical procedures. These findings emphasize the need to tailor healthcare interventions to cultural practices, which can help optimize resource allocation and improve healthcare delivery, even with limited resources.

## Supporting information

**S1 Checklist.  Inclusivity in global research.**
(DOCX)

## Acknowledgments

We extend our sincere gratitude to the Dhaka Community Hospital Trust for providing the human resources and materials necessary for the operation of our clinic, and to the staff of the same facility for their support in our clinical services.

## Author contributions

**Conceptualization:** Mototaka Inaba, Yui Yumiya, Tatsuhiko Kubo.

**Data curation:** Odgerel Chimed-Ochir.

**Formal analysis:** Odgerel Chimed-Ochir.

**Funding acquisition:** Taizo Sakata.

**Investigation:** Mototaka Inaba.

**Methodology:** Md Omar Sharif Ibne Hasan.

**Project administration:** Taizo Sakata, Md Omar Sharif Ibne Hasan, Mototaka Inaba.

**Supervision:** Yasuhiko Kamiya, Tatsuhiko Kubo.

**Writing – original draft:** Taizo Sakata, Odgerel Chimed-Ochir.

**Writing – review & editing:** Taizo Sakata, Yasuhiko Kamiya, Yui Yumiya, Tatsuhiko Kubo.

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
