## [Decision Letter · Decision Letter 0]

PGPH-D-24-00196

The effects of Ramadan on clinical consultation frequency in a Rohingya refugee camp, Bangladesh

Dear Dr. Sakata,

Thank you for submitting your manuscript to PLOS Global Public Health. After careful consideration, we feel that it has merit but does not fully meet PLOS Global Public Health’s publication criteria as it currently stands. Therefore, we invite you to submit a revised version of the manuscript that addresses the points raised during the review process.

We look forward to receiving your revised manuscript.

Kind regards,

Jianhong Zhou

Staff Editor

Journal Requirements:

1. We ask that a manuscript source file is provided at Revision. Please upload your manuscript file as a .doc, .docx, .rtf or .tex.

Additional Editor Comments (if provided):

Reviewers' comments:

Reviewer's Responses to Questions

**Comments to the Author**

1. Does this manuscript meet PLOS Global Public Health’s publication criteria ? Is the manuscript technically sound, and do the data support the conclusions? The manuscript must describe methodologically and ethically rigorous research with conclusions that are appropriately drawn based on the data presented.

Reviewer #1: Partly

Reviewer #2: Partly

2. Has the statistical analysis been performed appropriately and rigorously?

Reviewer #1: No

Reviewer #2: Yes

3. Have the authors made all data underlying the findings in their manuscript fully available (please refer to the Data Availability Statement at the start of the manuscript PDF file)?

Reviewer #1: No

Reviewer #2: No

4. Is the manuscript presented in an intelligible fashion and written in standard English?

Reviewer #1: No

Reviewer #2: Yes

5. Review Comments to the Author

Reviewer #1: Thank you very much for asking me to review this paper. I would agree there is a paucity of data on the topic of the effect of Ramadan on the health of refugees living in camps, thus this is a valid question and researching this a laudable effort.

I do have some concerns though about the presentation of the data and the author’s conclusions. This is essentially an extraction of quantitative data from a database that was not created for this research but for an administrative purpose. There is a complete lack of any qualitative information, either from the population served or the healthcare workers involved that could have informed the discussion, the latter which is based entirely on comparing the quantitative data, somewhat speculative, with those of other studies. Some of those other studies had at least a qualitative component.

Here some more specific comments:

• Is the study clinic the only one providing healthcare to the population 24h? Are there any other health service providers that could have biased the clinic attendance in this service? (E.g. in many camps there is a setup that out of hours care is provided by a different organisation).

• Some clearer description of the study design would be helpful, e.g. purpose, objectives, definition of study groups, definition of study type. It should be made clearer that this is a retrospective cohort study from a data collected for a more administrative purpose. The reason for selecting the five health events is not clearly described – was this because these were the most common? Or thought of the most relevant for Ramadan related health conditions?

• Whilst the number of datasets is large, the group sizes are highly discrepant. It would be helpful to provide a statistical power calculation for the expected/encountered outcomes.

• Why is there such a predominance of female attendees at the clinic? Does this reflect the camps demographics? Did you include children in the data, and what do you know about their adherence to Ramadan? Children carry a large proportion of gastrointestinal conditions. Did you analyse this to avoid potential bias?

• Having qualitative data from the same study population would be helpful to enlighten and give meaning to the data, rather than estimating the context from other studies. I appreciate this can be difficult in a camp situation. Alternatively, you could conduct focus groups with the health workers involved to inform your findings.

• The last paragraph of the discussion is a bit out of context, as you present no data around reproductive health. You may consider removing this or reframing this.

• The limitations should include a discussion about the limited interpretability of quantitative data.

• Figure 1 deserves more explanation – there are huge fluctuations in the number of health events throughout the year, whilst the change during Ramadan appears very small. Why is that? What events do the peaks and troughs represent?

Reviewer #2: The study is interesting and timely. However there are some points that need further clarification. While the current study focuses on the effects of Ramadan on clinical consultation frequency in the Rohingya refugee camp, it does not reference any additional literature that explores similar effects in diverse healthcare settings, and populations. Also, the authors state in the limitations section that the analysis includes patients from the host community, whose religion is not surveyed and may not be as consistent as the Rohingya population. This could introduce variability in the data and affect the generalizability of the findings. Ramadan is associated with a specific religious practice and denomination and the majority of Rohingya refugee population is Muslim so what is the point of including non-muslim population in the study? The inclusion of host population in the study is not sufficiently explained and documented. Limiting the sample to the Rohingya refugee population only would be more accurate, in line with the scope and objective of the study.

The authors state that the control period is from January 2020 to December 2020, excluding certain dates during Ramadan and other public holidays. However, they do not explicitly state whether the control period is before or after Ramadan. I suggest to specify this point, which may have an effect on the frequency of consultation and the types of health events reported. Time series analysis might offer a more nuanced understanding of the consultation patterns.

6. PLOS authors have the option to publish the peer review history of their article (what does this mean? ). If published, this will include your full peer review and any attached files.

**Do you want your identity to be public for this peer review?** For information about this choice, including consent withdrawal, please see our Privacy Policy .

Reviewer #1: No

Reviewer #2: No

---

## [Decision Letter · Decision Letter 1]

PGPH-D-24-00196R1

Effects of Ramadan on clinical visits at a primary health center in a Rohingya refugee camp in Bangladesh

Dear Dr. Sakata,

Thank you for submitting your manuscript to PLOS Global Public Health. After careful consideration, we feel that it has merit but does not fully meet PLOS Global Public Health’s publication criteria as it currently stands. Therefore, we invite you to submit a revised version of the manuscript that addresses the points raised during the review process.

The revied manuscript has been re-assessed by the reviewers and their comments are available below. Although the reviewers believe the revisions have strengthened the manuscript they still have some additional points that need to be addressed. Please review their comments and make the appropriate revisions to the manuscript

We look forward to receiving your revised manuscript.

Kind regards,

Emma Campbell, Ph.D

Staff Editor

Reviewers' comments:

Reviewer's Responses to Questions

**Comments to the Author**

1. If the authors have adequately addressed your comments raised in a previous round of review and you feel that this manuscript is now acceptable for publication, you may indicate that here to bypass the “Comments to the Author” section, enter your conflict of interest statement in the “Confidential to Editor” section, and submit your "Accept" recommendation.

Reviewer #1: (No Response)

Reviewer #2: All comments have been addressed

2. Does this manuscript meet PLOS Global Public Health’s publication criteria ? Is the manuscript technically sound, and do the data support the conclusions? The manuscript must describe methodologically and ethically rigorous research with conclusions that are appropriately drawn based on the data presented.

Reviewer #1: Partly

Reviewer #2: (No Response)

3. Has the statistical analysis been performed appropriately and rigorously?

Reviewer #1: Yes

Reviewer #2: (No Response)

4. Have the authors made all data underlying the findings in their manuscript fully available (please refer to the Data Availability Statement at the start of the manuscript PDF file)?

Reviewer #1: Yes

Reviewer #2: (No Response)

5. Is the manuscript presented in an intelligible fashion and written in standard English?

Reviewer #1: No

Reviewer #2: (No Response)

6. Review Comments to the Author

Reviewer #1: Dear Authors,

Thank you very much for addressing the various queries by the reviewers. I acknowledge that the manuscript is now more stronger, and heading towards the possibility of acceptance.

I have still some concerns though:

- I think you need to highlight very prominently, in the study design section as well as limitations, that your data were NOT collected for research or publication purposes, BUT for administrative reasons.

- I still struggle with your interpretation of the data which remains very speculative, in particular the section from line 468 where you provide no evidence that such reasons applied. I notice there were higher attendances for new appointments/routine health visits in the pre-Ramadan period as compared to the post Ramadan phase - could it be possible that patients were planning ahead bringing such appointments forward anticipating the holidays? Could you break these data down into weeks/months rather than 3 periods? Your numbers should allow for that. Please undertake a more systematic literature search to underpin your interpretations of your data, if possible from the same setting and time period, including 'grey literature' such as reports by organisations.

- Some minor points: Should it be 'sex' instead of 'gender'? I suggest to not make reference to 'Burma' because of the colonial connotations.

Reviewer #2: (No Response)

7. PLOS authors have the option to publish the peer review history of their article (what does this mean? ). If published, this will include your full peer review and any attached files.

**Do you want your identity to be public for this peer review?** For information about this choice, including consent withdrawal, please see our Privacy Policy .

Reviewer #1: No

Reviewer #2: No

---

## [Decision Letter · Decision Letter 2]

PGPH-D-24-00196R2

Effects of Ramadan on clinical visits at a primary health center in a Rohingya refugee camp in Bangladesh

Dear Dr. Sakata,

Thank you for submitting your manuscript to PLOS Global Public Health. After careful consideration, we feel that it has merit but does not fully meet PLOS Global Public Health’s publication criteria as it currently stands. Therefore, we invite you to submit a revised version of the manuscript that addresses the points raised during the review process.

I have checked the comments made by the reviewers in this round of revision as well as in your earlier versions. They have added important comments. Please address them carefully along with my comments below:

Why did you focus on certain conditions? Is this because the healthcare facility where you collected data provides only these services? The findings you generated for these particular conditions are very much understandable. It is very common for all Muslim people, including Muslims in a normal context, to receive lower healthcare services because they prioritize prayers and related issues. This time-related issue also often comes with religious concerns. For instance, checking for diabetes often involves getting a blood sample, which is not allowed during fasting according to religious beliefs. You recorded a higher number of arthritis patients because fasting can aggravate arthritis symptoms, and seeking treatment for it does not require anything that contradicts religious practices, such as taking a blood sample.I am a bit surprised to see the effects of fasting on post-natal care service access. What could be the reason for such higher odds, given that Rohingya women usually avoid post-natal care unless complications arise? Did you consider post-natal care only for those who delivered at the healthcare facilities where you collected data and received care before leaving the facilities?How many models did you run for Table 4? Was it just one? If so, how did you address multicollinearity? What factors were adjusted?

We look forward to receiving your revised manuscript.

Kind regards,

Nuruzzaman Khan, Ph.D.

Academic Editor

Additional Editor Comments (if provided):

Reviewers' comments:

Reviewer's Responses to Questions

**Comments to the Author**

1. If the authors have adequately addressed your comments raised in a previous round of review and you feel that this manuscript is now acceptable for publication, you may indicate that here to bypass the “Comments to the Author” section, enter your conflict of interest statement in the “Confidential to Editor” section, and submit your "Accept" recommendation.

Reviewer #1: All comments have been addressed

2. Does this manuscript meet PLOS Global Public Health’s publication criteria ? Is the manuscript technically sound, and do the data support the conclusions? The manuscript must describe methodologically and ethically rigorous research with conclusions that are appropriately drawn based on the data presented.

Reviewer #1: Yes

3. Has the statistical analysis been performed appropriately and rigorously?

Reviewer #1: Yes

4. Have the authors made all data underlying the findings in their manuscript fully available (please refer to the Data Availability Statement at the start of the manuscript PDF file)?

Reviewer #1: Yes

5. Is the manuscript presented in an intelligible fashion and written in standard English?

Reviewer #1: Yes

6. Review Comments to the Author

Reviewer #1: Thank you very much for addressing the remaining issues so comprehensively. I do believe that this has substantially enriched your analysis, discussion and conclusions in an area where research is sparse; and will provide questions and starting points for further research.

7. PLOS authors have the option to publish the peer review history of their article (what does this mean? ). If published, this will include your full peer review and any attached files.

**Do you want your identity to be public for this peer review?** For information about this choice, including consent withdrawal, please see our Privacy Policy .

Reviewer #1: No

---

## [Decision Letter · Decision Letter 3]

Effects of Ramadan on clinical visits at a primary health center in a Rohingya refugee camp in Bangladesh

PGPH-D-24-00196R3

Dear Dr Sakata,

We are pleased to inform you that your manuscript 'Effects of Ramadan on clinical visits at a primary health center in a Rohingya refugee camp in Bangladesh' has been provisionally accepted for publication in PLOS Global Public Health.

Best regards,

Julia Robinson

Executive Editor

Reviewer Comments (if any, and for reference):

Reviewer's Responses to Questions

**Comments to the Author**

1. If the authors have adequately addressed your comments raised in a previous round of review and you feel that this manuscript is now acceptable for publication, you may indicate that here to bypass the “Comments to the Author” section, enter your conflict of interest statement in the “Confidential to Editor” section, and submit your "Accept" recommendation.

Reviewer #1: All comments have been addressed

2. Does this manuscript meet PLOS Global Public Health’s publication criteria ? Is the manuscript technically sound, and do the data support the conclusions? The manuscript must describe methodologically and ethically rigorous research with conclusions that are appropriately drawn based on the data presented.

Reviewer #1: Yes

3. Has the statistical analysis been performed appropriately and rigorously?

Reviewer #1: Yes

4. Have the authors made all data underlying the findings in their manuscript fully available (please refer to the Data Availability Statement at the start of the manuscript PDF file)?

Reviewer #1: Yes

5. Is the manuscript presented in an intelligible fashion and written in standard English?

Reviewer #1: Yes

6. Review Comments to the Author

Reviewer #1: I will not make further comments to the authors as all requests came from another reviewer.

7. PLOS authors have the option to publish the peer review history of their article (what does this mean? ). If published, this will include your full peer review and any attached files.

**Do you want your identity to be public for this peer review?** For information about this choice, including consent withdrawal, please see our Privacy Policy .

Reviewer #1: No
